# Candidiasis in *Choloepus* sp.—A Review of New Advances on the Disease

**DOI:** 10.3390/ani14142092

**Published:** 2024-07-17

**Authors:** Marina Sette Camara Benarrós, Felipe Masiero Salvarani

**Affiliations:** Instituto de Medicina Veterinária, Universidade Federal do Pará, Castanhal 68740-970, PA, Brazil; marina7camara@gmail.com

**Keywords:** fungus, real sloth, gastrointestinal tract, disease, wild mammal

## Abstract

**Simple Summary:**

Candidiasis is a fungal infection caused by *Candida* species, which can affect various animals, including *Choloepus* species (two-toed sloths). In these sloths, candidiasis typically occurs when their immune system is weakened, allowing *Candida* fungi to overgrow and cause infection. The disease can manifest in different parts of the body, such as the mouth, skin, and digestive tract, leading to symptoms like lesions, inflammation, and discomfort. Factors that can predispose *Choloepus* sp. to candidiasis include stress, poor nutrition, and other underlying health conditions. Diagnosis usually involves identifying the presence of *Candida* fungi through laboratory tests. Treatment typically includes antifungal medications and supportive care to address any underlying health issues and improve the sloth’s overall condition. Understanding candidiasis in *Choloepus* sp. is essential for ensuring the health and well-being of these animals, especially in captivity where they might be more susceptible to infections.

**Abstract:**

Candidiasis is a significant fungal infection caused by various species of the genus *Candida*, posing health challenges to a wide range of animals, including *Choloepus* species (two-toed sloths). This review article aims to provide a comprehensive understanding of candidiasis in *Choloepus* sp., highlighting the etiology, epidemiology, pathogenesis, clinical manifestations, diagnosis, treatment, and prevention strategies. This article begins by examining the causative agents, primarily focusing on *Candida albicans*, which is the most commonly implicated species in candidiasis. The epidemiological aspects are discussed, emphasizing the prevalence of candidiasis in wild and captive *Choloepus* populations and identifying predisposing factors, such as immunosuppression, stress, poor nutrition, and environmental conditions. Pathogenesis is explored, detailing the mechanisms through which *Candida* species invade host tissues and evade immune responses. Clinical manifestations in *Choloepus* sp. are described, including oral thrush, cutaneous lesions, and gastrointestinal infections, and their impact on the health and behavior of affected individuals. Diagnostic methods, including culture techniques, histopathology, and molecular assays, are reviewed to highlight their roles in accurately identifying *Candida* infections. This article also covers treatment options, focusing on antifungal therapies and supportive care tailored to the unique physiology of *Choloepus* sp. Finally, prevention and management strategies are discussed, emphasizing the importance of maintaining optimal husbandry practices, regular health monitoring, and early intervention to reduce the incidence and impact of candidiasis in *Choloepus* populations. This review underscores the need for further research to enhance our understanding of candidiasis and improve health outcomes for these unique and vulnerable animals.

## 1. Introduction

Candidiasis is a notable fungal infection caused by opportunistic yeasts belonging to the genus *Candida*. These fungi are commonly found in the environment and as part of the normal flora of various animal species. Under certain conditions, however, *Candida* species can overgrow and lead to infection, resulting in significant health issues. Among the various animals affected by candidiasis, *Choloepus* species (two-toed sloths) have garnered attention due to their unique physiology and the increasing number of cases reported, particularly in captive environments [1,2,3,4].

*Choloepus* species, comprising *Choloepus hoffmanni* and *Choloepus didactylus,* are arboreal mammals native to Central and South America. These sloths are characterized by their slow metabolism, specialized diet, and distinctive behaviors, all of which contribute to their unique health care needs. The rise in popularity of sloths in captivity, whether in zoos, rescue centers, or as exotic pets, has led to increased scrutiny of their health and well-being. Among the health challenges faced by captive sloths, candidiasis has emerged as a significant concern [1,4].

Candidiasis in *Choloepus* sp. can manifest in various forms, including oral thrush, cutaneous lesions, and gastrointestinal infections. These infections can compromise the overall health of the sloths, leading to weight loss, discomfort, and secondary infections. Factors such as stress, poor nutrition, compromised immune function, and inadequate husbandry practices are often implicated as predisposing factors for the development of candidiasis in these animals [5,6].

The aim of this review is to provide a comprehensive overview of candidiasis in *Choloepus* sp., encompassing the etiology, epidemiology, pathogenesis, clinical manifestations, diagnosis, treatment, and prevention of this infection. By synthesizing current knowledge and identifying gaps in the research, this review seeks to enhance our understanding of candidiasis in two-toed sloths and improve strategies for their care and management in both captive and wild settings. Through a detailed exploration of this disease, we aim to contribute to the growing body of veterinary literature focused on the health and conservation of these unique and captivating animals.

## 2. *Candida*

The genus *Candida* sp. is composed of eukaryotic yeasts that lack photosynthetic pigments, are asexual and dimorphic, and are covered by a cell wall rich in chitin, mannans, glucans and of the plasma membrane rich in ergosterol, all using carbon dioxide (CO_2_) (however, at beneficial physiological levels, as high levels of CO_2_ inhibit the growth of *Candida*) absorbed from the environment as their main energy source. The cell wall of *Candida* species is rich in chitin and ergosterol, which are key to maintaining their structural integrity and resistance to environmental stress. *Candida* species primarily reproduce asexually through budding, although some species can also undergo sexual reproduction. The dimorphic nature of these fungi (some species of *Candida* are dimorphic, not all) allows them to switch between yeast and filamentous forms, enhancing their ability to invade host tissues and evade the immune response [1,2,7] (Figure 1).

*Candida* species are facultative anaerobes capable of utilizing various carbon sources, including glucose and, uniquely, CO_2_ absorbed from the environment (however, at beneficial physiological levels, as high levels of CO_2_ inhibit the growth of *Candida*). This metabolic flexibility allows them to thrive in diverse environments, from mucosal surfaces to the bloodstream. They are considered commensal fungi that naturally inhabit the mucosal microbiota of animals and humans, with more than 200 species already described [1,2,3,4]. However, there are increasing reports of infections by *Candida* sp. linked to predisposing factors such as age, indiscriminate use of medications, and immunosuppression of individuals, with the most affected anatomical sites being the integument, urinary, gastrointestinal, and reproductive tracts [7].

The main species considered zoonotic and associated with infection in domestic and wild animals are *Candida albicans*, *C. tropicalis*, *C. parapsilosis*, and *C. guilliermondii*, with the first responsible for 60% of isolations in humans due to disruptions in the parasite–host balance facilitating the adherence and multiplication of the agents. Consequently, these yeasts have become an important public health problem due to the high resistance of the agents to conventional treatments and difficulties in isolation and diagnosis [3,4,8]. The distribution of Candida species shows regional variations. For instance, *C. albicans* is universally prevalent, while *C. tropicalis* is more common in tropical and subtropical regions. *C. glabrata* and *C. krusei* have been increasingly reported in Europe and North America, reflecting changes in antifungal resistance patterns [8].

The prevalence of *Candida* infections varies globally, with higher rates observed in hospital settings due to invasive procedures and the use of broad-spectrum antibiotics. In the United States and Europe, *Candida* species are a leading cause of nosocomial infections, particularly in immunocompromised patients. In developing countries, the burden of *Candida* infections is also significant, often exacerbated by limited access to healthcare and diagnostic facilities [3,4,9].

Antifungal resistance is a growing concern worldwide. *C. glabrata* and *C. krusei* exhibit high levels of resistance to azole antifungals, complicating treatment strategies. The emergence of multidrug-resistant strains, particularly in hospital settings, underscores the need for ongoing surveillance and the development of new antifungal agents [1,10].

*Candida* species are versatile and adaptive pathogens with significant clinical and public health implications. Understanding their microbiological characteristics, species diversity, and global occurrence is crucial for developing effective prevention and treatment strategies. Continued research and surveillance are essential to address the challenges posed by antifungal resistance and to improve outcomes for individuals affected by *Candida* infections. This review highlights the importance of a comprehensive approach to studying *Candida*, integrating microbiological insights with epidemiological data to inform better clinical practices and public health policies [2,3,4,10].

## 3. Epidemiology

Regarding wild animals, the significant increase in emerging fungal diseases has affected the survival of certain populations, favoring the extinction process of more susceptible species and the greater spread of zoonoses. Therefore, investing in efficient diagnostic techniques also becomes essential for the maintenance of individuals both in the wild and in captivity, in addition to favoring the understanding of diseases that are still poorly reported in terms of clinical and health aspects [2,3,4].

The presence of fungi with high zoonotic potential has already been described in this superorder, requiring attention due to their potential capacity to develop diseases in humans, such as *Paracoccidioides brasiliensis*, detected in organ samples like the liver and spleen of armadillos (*Dasypus novemcinctus*) examined in the Tucuruí region, Pará, Brazil. These data are extremely important for understanding the ecology of fungi due to the high consumption of the meat and viscera of this species by native communities [11].

*Candida* sp. has been described in bears, dolphins, wild felines, primates, birds, and xenarthrans as one of the main pathogenic fungi of importance in captivity, mainly affecting the gastrointestinal tract of young animals. The primary mode of transmission is direct contact or contact with surfaces containing contaminated feces, which underscores the importance of quarantines, use of personal protection when handling materials and animals, and thorough cleaning of enclosures and utensils due to the high zoonotic potential of the agents [12,13].

Candidiasis is also frequently associated with morbidity and mortality in birds due to impairments in the normal intestinal microbiota. Generally, these individuals can maintain yeasts in non-pathogenic quantities in the gastrointestinal tract, but imbalances associated with stress, indiscriminate use of antibiotics, and nutritional deficiencies favor the propagation and dissemination of *Candida*, which becomes pathogenic, causing serious systemic damage to individuals that can lead to death [2,3,4,12,13].

## 4. Pathogenesis

The pathogenesis of candidiasis in *Choloepus* species is multifactorial, involving interactions between the host’s immune system, the virulence of Candida species, and environmental conditions. In healthy sloths, *Candida* species are typically commensal organisms residing on the mucosal surfaces of the gastrointestinal tract, skin, and other areas without causing disease. However, when the host’s immune defenses are compromised, these fungi can become opportunistic pathogens [2,3,4,14].

Several factors can predispose *Choloepus* sp. to candidiasis, including stress, poor nutrition, concurrent infections, and immunosuppressive conditions. Stress, which can arise from environmental changes, captivity, or handling, can weaken the immune response, allowing *Candida* species to overgrow. Nutritional deficiencies, particularly in vitamins and minerals, can also impair the immune system, making sloths more susceptible to infections [3,4,14,15].

*Candida* species exhibit several virulence factors that enhance their pathogenicity. These include the ability to switch between yeast and hyphal forms, the secretion of hydrolytic enzymes like proteases and phospholipases, and the formation of biofilms. The yeast-to-hyphal transition is crucial for tissue invasion and immune evasion, while hydrolytic enzymes facilitate the breakdown of host tissues and promote dissemination. Biofilm formation on mucosal surfaces and medical devices further complicates treatment by providing a protective environment against antifungal agents and the host immune system [3,4,16,17].

Once *Candida* invades the host tissues, it triggers an inflammatory response. The fungi adhere to epithelial cells, invade the mucosal barrier, and proliferate, causing localized or systemic infections. In *Choloepus* sp., candidiasis can affect various organs, leading to different clinical manifestations depending on the site of infection [3,4,18,19].

## 5. Clinical Manifestations

The clinical manifestations of candidiasis in *Choloepus* sp. are diverse and depend on the site and severity of the infection. In many cases, the disease presents with nonspecific symptoms, making diagnosis challenging without laboratory confirmation. Common clinical signs include weight loss, lethargy, and poor coat condition, which may be indicative of systemic infection or underlying health issues [2,3,4,14,15].

Oral candidiasis, or thrush, is one of the more recognizable forms of the disease, presenting as white, curd-like plaques on the mucous membranes of the mouth and tongue. Affected sloths may exhibit difficulty eating, drooling, and oral discomfort. Cutaneous candidiasis can present as red, inflamed skin lesions, often in areas subjected to moisture and friction, such as skin folds and the perineal area [7,12].

Gastrointestinal candidiasis is particularly concerning in young sloths, where it can lead to severe digestive disturbances. Symptoms include diarrhea, bloating, and anorexia. In severe cases, the infection can spread to the bloodstream, leading to systemic candidiasis or candidemia. This disseminated form of the disease can affect multiple organs, including the liver, spleen, kidneys, and lungs, and is often fatal if not promptly treated [13].

Respiratory candidiasis, though less common, can occur in immunocompromised individuals or those with underlying respiratory conditions. Signs include nasal discharge, coughing, and difficulty breathing. The presence of *Candida* species in the respiratory tract can exacerbate pre-existing conditions and lead to secondary bacterial infections [3,4,14].

The varied clinical presentations of candidiasis in *Choloepus* sp. necessitate a thorough clinical examination and the use of diagnostic tools such as culture, histopathology, and molecular techniques to confirm the presence of *Candida* and determine the appropriate treatment. Early detection and intervention are critical to managing the infection and improving the prognosis for affected sloths [3,4,14,20].

Most wild species exhibit symptoms, including inappetence, emesis, diarrhea, skin lesions, and lethargy. Diagnosis is similar to that performed in small animals, making the correct identification of the agent crucial. Different *Candida* species affect these individuals, facilitating specific treatment and avoiding the development of drug resistance and the death of very sensitive animals [3,4,14,15,19,20].

## 6. Diagnostic and Treatment

Diagnosing candidiasis in *Choloepus* species (two-toed sloths) requires a combination of clinical observation, laboratory tests, and advanced diagnostic techniques. Given the nonspecific nature of clinical symptoms, the accurate identification of *Candida* infections is essential for effective treatment and management [21]. The diagnostic process begins with a thorough clinical examination, noting any signs, such as oral plaques, cutaneous lesions, gastrointestinal disturbances, or respiratory symptoms. Because these signs are not unique to candidiasis, further diagnostic tests are necessary to confirm the presence of *Candida* species [22].

One of the primary diagnostic tools is the culture of samples from affected tissues or bodily fluids. Samples can be taken from oral swabs, skin scrapings, fecal matter, or blood, depending on the suspected site of infection. These samples are cultured on Sabouraud dextrose agar or chromogenic media to encourage the growth of *Candida* colonies. Positive cultures are followed by microscopic examination to identify the characteristic budding yeast cells and pseudohyphae, which are indicative of *Candida* species [21,22].

Histopathological examination of biopsied tissues provides additional confirmation. Tissue samples are stained with periodic acid-Schiff (PAS) or Gomori methenamine silver (GMS) stains, which highlight the fungal elements within the tissue. This method helps to visualize the extent of tissue invasion and inflammation, providing insights into the severity of the infection [22].

Molecular diagnostic techniques, such as polymerase chain reaction (PCR), offer high sensitivity and specificity for detecting *Candida* DNA. This method involves the amplification of specific DNA sequences, which allows for the precise detection and differentiation of *Candida* species. For instance, Nadăș et al. [23] conducted a comparative study using phenotypic and PCR-RFLP (restriction fragment length polymorphism) methods to identify *Candida* species isolated from animals. They designed specific primers targeting the internal transcribed spacer (ITS) regions of the ribosomal RNA gene complex, which are highly variable among different *Candida* species. The primers used in their study were ITS1: 5′-TCC GTA GGT GAA CCT GCG G-3′ and ITS4: 5′-TCC TCC GCT TAT TGA TAT GC-3′. These primers amplify a region approximately 500–800 base pairs long, depending on the *Candida* species. The resulting amplicons can then be digested with restriction enzymes to generate species-specific patterns in PCR-RFLP analysis. By employing these primers and analyzing the resulting amplicons, researchers can accurately identify and differentiate between various *Candida* species present in clinical samples from *Choloepus* sp. This method enhances the diagnostic accuracy, allowing for appropriate treatment strategies to be developed. Example amplicons for common *Candida* species include the following: *Candida albicans* (approximately 530 bp), *Candida glabrata* (approximately 800 bp), *Candida parapsilosis* (approximately 580 bp), and *Candida tropicalis* (approximately 600 bp). Incorporating these specific primers and detailing the expected amplicon sizes can significantly improve the clarity and utility of the PCR, providing readers with concrete examples and a clear understanding of how this diagnostic method is applied in the context of Candidiasis in *Choloepus* sp. These methods are particularly useful in cases where traditional culture methods may be inconclusive or when rapid diagnosis is required [21,23].

Serological tests can also aid in diagnosing candidiasis by detecting antibodies or antigens associated with *Candida* infections. However, these tests are less commonly used in veterinary practice and may have limited specificity due to cross-reactivity with other fungal pathogens [3,4,21,22,23,24].

The effective treatment of candidiasis in *Choloepus* sp. involves a combination of antifungal therapy, supportive care, and environmental management. The choice of antifungal agent and treatment duration depends on the severity and location of the infection, as well as the specific *Candida* species involved [14,25].

Azole antifungals, such as fluconazole (5–10 mg/kg orally or intravenously every 24 h) and itraconazole (5 mg/kg orally every 24 h), are commonly used to treat candidiasis in sloths. These drugs inhibit ergosterol synthesis, disrupting the fungal cell membrane integrity. For systemic infections, fluconazole is often preferred due to its excellent oral bioavailability and ability to penetrate various tissues. Itraconazole may be used for refractory cases or when a broader spectrum of antifungal activity is needed [14,26,27,28,29,30].

Echinocandins, such as caspofungin (1 mg/kg intravenously every 24 h), are another class of antifungals that inhibit the synthesis of β-(1,3)-D-glucan, an essential component of the fungal cell wall. These agents are particularly effective against *Candida* species that exhibit resistance to azoles. However, their use in veterinary medicine is less common due to cost and availability. For topical infections, nystatin (100,000 IU/kg orally every 12 h) or clotrimazole (topical application, 2–3 times daily) can be applied directly to the affected areas. These antifungals are effective for treating oral thrush and cutaneous lesions by disrupting the fungal cell membrane. Supportive care plays a crucial role in the treatment of candidiasis. Ensuring proper nutrition, hydration, and stress reduction is essential for the recovery of affected sloths. Providing a balanced diet rich in essential nutrients can help bolster the immune system and promote healing. In cases of gastrointestinal candidiasis, probiotics may be administered to restore healthy gut flora and prevent further fungal overgrowth. Environmental management is critical to prevent the recurrence of candidiasis. This includes maintaining clean and dry living conditions, reducing environmental stressors, and implementing proper sanitation protocols. Regular cleaning and disinfection of enclosures, feeding utensils, and water sources can minimize the risk of reinfection. Here are some examples of each class of disinfectants commonly used against *Candida* species, such as alcohols (ethanol and isopropanol), phenolics (phenol, triclosan, and chlorhexidine), quaternary ammonium compounds (QACs) (benzalkonium chloride and cetylpyridinium chloride), halogens (chlorine, iodine, and sodium hypochlorite), hydrogen peroxide and peracetic acid, aldehydes (glutaraldehyde and formaldehyde), biguanides (chlorhexidine and polyaminopropyl biguanide), and heavy metals (silver nitrate and mercuric chloride). These disinfectants are selected based on their efficacy, safety, and the specific application environment. For instance, in clinical settings where *Candida* infections are a concern, a combination of alcohol-based hand sanitizers, surface disinfectants containing QACs or phenolics, and antiseptic solutions like chlorhexidine can be used to minimize the risk of transmission and infection [14,25,26,27,28,29,30].

## 7. Prevention and Control

Effective prevention and control strategies for candidiasis in *Choloepus* species (two-toed sloths) are crucial to maintaining the health of these animals, especially in captive environments. These strategies involve a combination of environmental management, nutritional support, stress reduction, and routine health monitoring [31].

Maintaining a clean and hygienic environment is fundamental to preventing candidiasis. Enclosures should be kept clean and dry to minimize the growth and spread of fungal spores. Regular cleaning protocols should include the thorough disinfection of surfaces, feeding utensils, and water sources using antifungal agents that are safe for use around animals. Proper ventilation within enclosures helps reduce humidity levels, which is important as *Candida* species thrive in moist conditions. Additionally, bedding materials should be regularly changed and kept dry to prevent fungal colonization [1,2,3,4].

Providing a balanced and nutritionally adequate diet is essential for the prevention of candidiasis. Sloths require a diet rich in vitamins, minerals, and other essential nutrients to support a robust immune system. Nutritional deficiencies can compromise the immune response, making sloths more susceptible to infections. Diets should be tailored to meet the specific needs of *Choloepus* sp., ensuring they receive adequate fiber, protein, and micronutrients. Supplementation with probiotics can also be beneficial in maintaining a healthy gut microbiota, which plays a role in preventing fungal overgrowth [1,2,3,4].

Stress is a significant predisposing factor for candidiasis in sloths. Stressful conditions, such as frequent handling, inadequate social interactions, and sudden changes in the environment, can weaken the immune system and increase susceptibility to infections. To mitigate stress, enclosures should mimic the natural habitat as closely as possible, providing ample climbing structures, foliage, and hiding spots. Caregivers should minimize handling and ensure that interactions with the sloths are calm and gentle. Additionally, efforts should be made to maintain a stable and predictable environment, avoiding sudden changes that could stress the animals [31].

Regular health monitoring and early detection are key components of candidiasis prevention. Routine veterinary check-ups, including physical examinations and laboratory tests, can help identify early signs of infection before they become severe. Monitoring weight, appetite, and behavior can provide valuable insights into the health status of the sloths. Any deviations from normal should prompt further investigation, including cultures and molecular diagnostics, to rule out candidiasis or other infections [1,2,3,4].

Implementing quarantine protocols for new or symptomatic animals is essential to prevent the spread of candidiasis within captive populations. Newly acquired sloths should be quarantined for a period sufficient to monitor for signs of infection and to perform necessary health screenings. Symptomatic animals should be isolated from the healthy population to prevent cross-contamination. Caregivers should use personal protective equipment (PPE) when handling quarantined or isolated animals to reduce the risk of zoonotic transmission and contamination [31].

Educating caregivers and staff about the importance of hygiene, stress reduction, and proper animal handling techniques is vital for the prevention and control of candidiasis. Training programs should cover the identification of early signs of infection, the implementation of cleaning and disinfection protocols, and the proper use of PPE. Informed and vigilant caregivers are better equipped to maintain a healthy environment and respond promptly to potential health issues [31].

By integrating these prevention and control measures, the incidence of candidiasis in *Choloepus* sp. can be significantly reduced. These strategies not only improve the overall health and well-being of captive sloths but also contribute to the broader efforts in wildlife conservation and management [14,31].

## 8. *Choloepus* sp.

### 8.1. Species Details

The two-toed sloths of the genus *Choloepus* belong to the superorder *Xenarthra*, order *Pilosa*, and family *Megalonychidae*. There are two species, *Choloepus didactylus* [15] and *Choloepus hoffmanni* [32], which are mainly differentiated by the lighter color of the neck fur in *C. hoffmanni* (Figure 2).

These are placental mammals known for their lethargic habits and low-energy diet consisting of leaves and fruits. Two-toed sloths can reach up to 86 cm in length and 8.4 kg in weight when fully grown. They are primarily distributed in the tropical forests from Venezuela to Peru and throughout the Brazilian Amazon [6,33,34].

### 8.2. Commonly Reported Diseases of Sloths from Captivity

Due to intense habitat fragmentation, the presence of sloths, armadillos, and anteaters in captivity is increasing due to various human-associated problems, such as road accidents, electric shocks, and burns. In captivity, these species become more susceptible to developing infectious diseases, either due to cross-contamination, the presence of comorbidities that make the animal more vulnerable, or close contact with pathogens in the hospital environment [14,35].

Diseases affecting captive sloths vary according to several factors, such as the animal’s age and clinical condition upon arrival, management, nutrition, and whether it has had contact with other individuals and potential anthropogenic actions (road accidents, burns from fires, or electric shocks) [16]. Most problems are observed within the first six months of captivity, especially in young or highly debilitated animals, and are related to digestive disorders, including dental alterations and nutritional deficiencies, followed by respiratory problems and traumatic injuries [17,18].

Gastrointestinal tract diseases are among the leading causes of death in captive sloths, partly due to the particularities associated with this system, such as the presence of multiple fermentation chambers adapted to a highly fibrous and low-calorie diet, dietary specifics that are difficult to replicate in captivity, or high sensitivity to dietary variability [19]. The presence of food-borne pathogens or those in the hospital environment and contact with other animals also promotes greater spread of commensal agents, primarily from the gastrointestinal and integumentary systems, associated with capture and daily handling stress and comorbidities, increasing the possibility of secondary infections [36].

Among infectious diseases, notable ones are enteritis caused by *Escherichia coli*, *Salmonella* sp., and *Aeromonas* sp. [17] and vesicular stomatitis caused by viruses and fungi of the genus *Candida* sp. and *Histoplasma* sp. [20]. The main symptoms include diarrhea, mucus in the feces, loss of appetite, tympanism, and colic, which can lead to death due to severe dysbiosis and associated pain [37].

The presence of trauma and burns also predisposes to alterations in the normal microbiota of the integument and pathogens, similar to what occurs in humans, further associated with various temperature and humidity conditions of the enclosures in most institutions that keep these animals, which is not very suitable for the species’ particularities [38]. Small carbon pigment foci and abundant fungal structures resembling *Candida* sp. have already been identified on the skin of anteaters injured by electric shocks (Figure 3), with the alterations still poorly described in the literature [14].

The main prophylactic methods are associated with food hygiene and the appropriate use of diets according to the biological demands of the individuals, as well as early diagnosis in the presence of any behavioral change or symptom. However, keeping xenarthrans, especially sloths, in captivity remains a challenge, mainly due to the difficulty in replicating the biological and nutritional demands that the species present when taken from the wild [14,39,40].

In the presence of clinical signs indicative of disease and potential diagnosis, targeted treatment of the causative agent and other comorbidities affecting the immune response is essential for the animal’s recovery, linked to sanitary, nutritional, and overall environmental management, ensuring the maintenance of the species in captivity [16,25].

### 8.3. Candidiasis in Sloths

Isolations of fungal and bacterial agents in xenarthrans are still scarce, but with the advancement of urbanization and the increased presence of these individuals in captivity, knowledge about these agents, especially those of zoonotic nature, becomes essential [40].

Regarding the fungal agents found in sloths, most are primarily of digestive and/or integumentary origin. The first isolation and identification of *Microsporum canis*, a dermatophyte fungus characteristically causing proliferative alopecia, was described in a sloth (*Bradypus tridactylus*) received at the Emílio Goeldi Museum in Belém, PA, Brazil [41].

Similarly, the same fungus and *Microsporum gypseum* were isolated for the first time in three specimens of the common sloth (*Bradypus variegatus*) from the state of Pernambuco, which also exhibited areas of alopecia and crusts adhered to the fur (Figure 4). In this case, the diagnosis was made from samples collected from the hair and crusts subjected to direct microscopic examination with 30% KOH and culture on Mycosel Agar. A direct examination revealed arthrospores in the hairs, and seven days after the culture, colonies suggestive of the genus *Microsporum* were observed, confirmed by the observation of macroconidia structures [42].

Regarding the genus *Choloepus* sp., there are reports of the occurrence of *Malassezia* sp. (Figure 5) and *Microsporum gypseum* [16] as causes of integumentary and auricular alterations such as alopecia, pruritus, and crust formation and *Paracoccidioides brasiliensis* [43], *Histoplasma capsulatum* [37], and *Candida* sp. [20] as causes of gastrointestinal alterations (gastric ulcers, mucosal desquamation, and dysbiosis).

Regarding *Candida* sp., the agent was isolated from a two-toed sloth (*C. hoffmanni*) with a history of bronchopneumonia due to gastric reflux, following ineffective treatments. A necropsy of the individual revealed a dilated distal rectum (Figure 6), gastric mucosa with multiple crater-like nodular formations, and pulmonary alterations indicative of necrosis [20].

Histopathologically, the squamous epithelium in the prepyloric region exhibited multiple areas of necrosis affecting the muscularis mucosae, along with cellular debris diagnosed as yeast and hyphae within the epithelium (Figure 7). This confirmed the diagnosis of multifocal necrotic fungal gastritis caused by *Candida* sp., which led to the animal’s death following bronchoaspiration [20].

Similarly, gastrointestinal alterations have been identified that promote stomatitis, gastritis, constipation, and enteritis as the primary causes of the highest morbidity and mortality rates in adult sloths (between 4 and 15 years) of the genus *Choloepus* sp. in North America, with the most severe cases involving stomatitis and ulcerative gastritis associated with *Candida* sp., diagnosed through cytology and oral swabs of the affected animals [20].

Regarding therapeutics, there are mentions of using itraconazole or ketoconazole, both at a dose of 5 mg/kg, orally, for 4 to 6 weeks for the treatment of disseminated dermatophytosis, accompanied by chlorhexidine baths to control the infection. Miconazole can be used topically for fungal otitis and terbinafine for the topical treatment of crusts present on the skin. All treatments require long periods of use of at least one to two weeks for real control of the infectious agent, emphasizing that it is important to associate supportive therapy with the supplementation and treatment of concomitant diseases for greater success in controlling fungi. There are still no studies addressing other systemic therapies for the treatment of *Candida* sp. in these animals, and it is frequently found in necropsies [16].

There is a notable need for studies with a greater focus on sloths and the occurrence of infectious alterations, as well as proper diagnosis and therapeutics, which directly impact these individuals in captivity, causing severe health damage that can even lead to premature death.

## 9. Conclusions

Candidiasis in *Choloepus* sp. represents a significant health concern, especially for individuals in captivity, where the prevalence of infectious diseases tends to increase due to various stressors and environmental factors. The unique biological and nutritional requirements of two-toed sloths make them particularly susceptible to gastrointestinal and integumentary infections, including those caused by *Candida* sp. The complexity of their digestive system, which includes multiple fermentation chambers adapted to a fibrous and low-calorie diet, poses challenges for maintaining their health in captivity. The occurrence of *Candida* infections in sloths has been associated with several clinical manifestations, including stomatitis, gastritis, constipation, and enteritis. These infections can lead to severe systemic issues and high mortality rates if not properly managed. The isolation of *Candida* sp. and other zoonotic fungal pathogens from sloths highlights the importance of comprehensive diagnostic approaches and the need for effective therapeutic protocols. The treatment of *Candida* infections in sloths often involves the use of antifungal medications such as itraconazole and ketoconazole, as well as supportive care to address underlying comorbidities and stress factors. However, the lack of specific studies on systemic therapies for *Candida* in sloths underscores the need for further research to develop tailored treatment strategies. Preventive measures, including strict hygiene practices, appropriate dietary management, and the early diagnosis of behavioral changes or symptoms, are crucial for minimizing the risk of infection. The increasing urbanization and habitat fragmentation necessitate greater attention to the health management of captive sloths to ensure their well-being and conservation. Overall, this study of candidiasis in *Choloepus* sp. reveals significant gaps in our understanding of the disease’s pathogenesis, diagnosis, and treatment. Addressing these gaps through dedicated research and improved veterinary practices will be essential for enhancing the care and survival of these unique and vulnerable animals in captivity.

## Figures and Tables

**Figure 1 animals-14-02092-f001:**
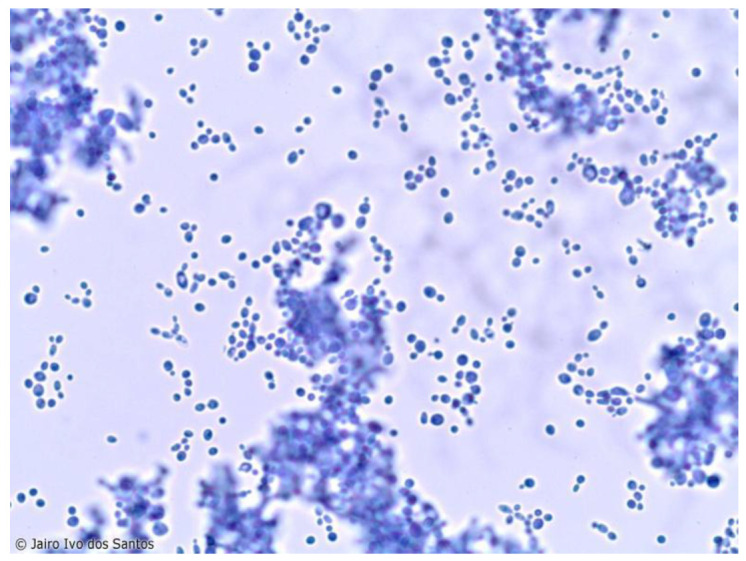
A single budding blastoconidia of *Candida* sp. present in the oral cavity of mammals (the use of Figure 1 has been authorized by Prof. Dr. Jairo Ivo dos Santos, intellectual owner of the same).

**Figure 2 animals-14-02092-f002:**
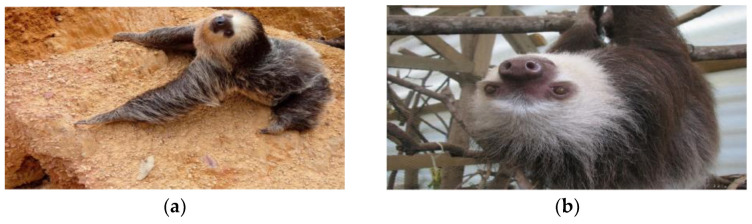
Specimens of sloth species from the *Megalonychidae* family. (**a**) *Choloepus didactylus* [15]. (**b**) *Choloepus hoffmanni* [32].

**Figure 3 animals-14-02092-f003:**
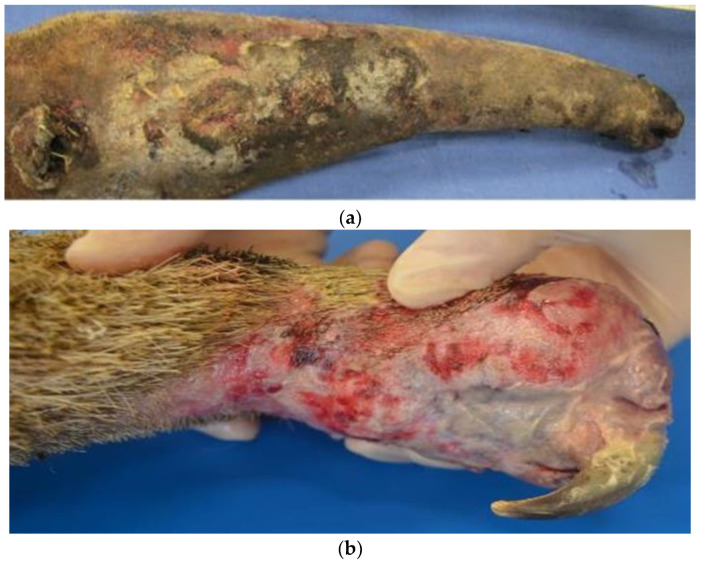
Giant anteaters with macroscopic burn lesions. (**a**) Focally extensive area of ulceration on the skin of the face, resulting from the charring and dissection of the burn tissues on the face. (**b**) The right thoracic limb with a focally extensive area of an integumentary burn with areas of granulation tissue and the loss of multiple digits [39].

**Figure 4 animals-14-02092-f004:**
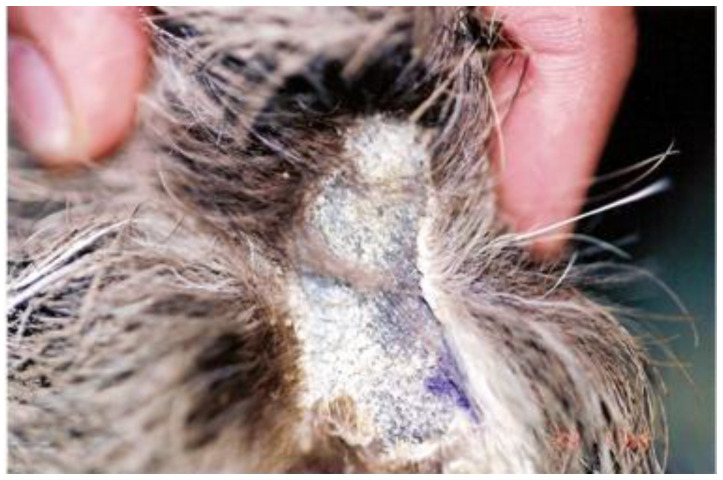
Dermatophytosis in *Bradypus variegatus*: alopecic area with circumscribed scaling on the pelvic limb [42].

**Figure 5 animals-14-02092-f005:**
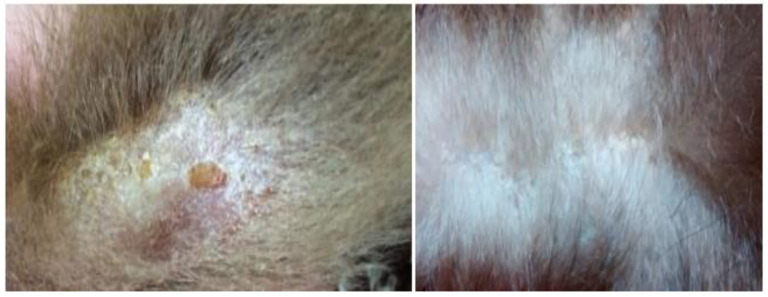
*Malassezia* sp. causing desquamation in the dermis of *Choloepus* sp. [16].

**Figure 6 animals-14-02092-f006:**
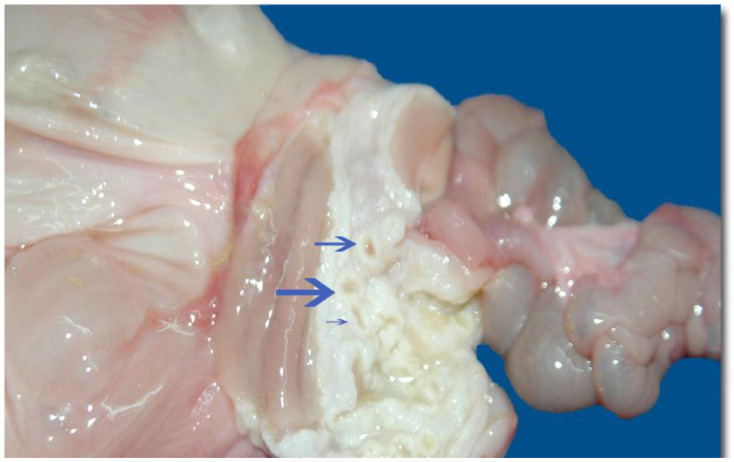
Prepyloric gastric squamous mucosa showing multiple volcanic crater-like growths in arrows. (Intellectual property figure of Alexis Berrocal. Histopatovet, Private Pathology Laboratory, Heredia, Costa Rica. Available in https://www.histopatovet.com/wp-content/uploads/2017/09/Sloth-Gastric-Candidiasis-Dallas.pdf) (accessed on 30 April 2024).

**Figure 7 animals-14-02092-f007:**
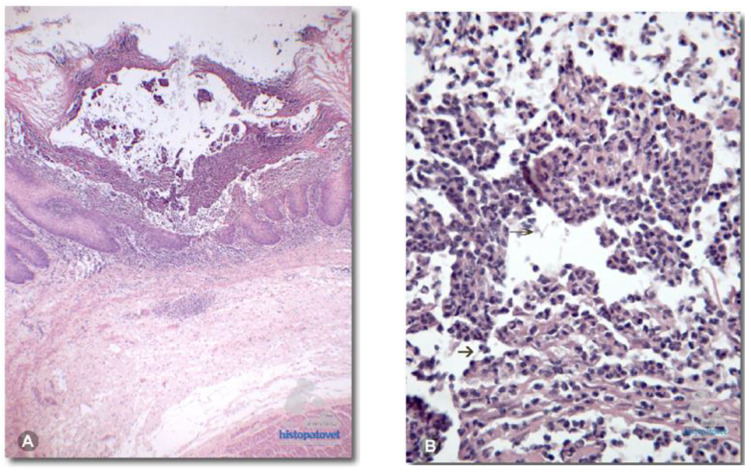
(**A**). The non-glandular mucosa of the prepyloric stomach presents multiple necrotic centers reaching the muscular layer of the mucosa. (**B**). The presence of necrotic debris. (Intellectual property figure of Alexis Berrocal. Histopatovet, Private Pathology Laboratory, Heredia, Costa Rica. Available in https://www.histopatovet.com/wp-content/uploads/2017/09/Sloth-Gastric-Candidiasis-Dallas.pdf) (accessed on 30 April 2024).

## Data Availability

The original contributions presented in the study are included in the article, further inquiries can be directed to the corresponding author.

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
