# Peer review of "Candidiasis in Choloepus sp.—A Review of New Advances on the Disease"

_animals, 2024, doi:10.3390/ani14142092_

Round 1
Reviewer 1 Report
Comments and Suggestions for Authors
The present manuscript is a valuable one, providing details about the lesional, therapeutic, prophylactic aspects of candidiasis, in an animal species less studied by researchers. This article can help practicing veterinarians in combating this fungal disease in captive sloths. The manuscript is well written in English, thoroughly structured, but nevertheless there are some aspects that can be improved before being accepted for publication. These can be found below:
Line 12: Please add space before “Factors…”
Line 16: Please use Italics for “Choloepus”
Line 19: Please use Italics for “Candida”
Line 28: Please use Italics for “Candida”
Line 148: Please use Italics for “Candida”
Line 195: Please remove “,” after 20….. from “[3,4,14,20,]”
Lines 221-225: The paragraph related to the PCR method is too general, it does not provide clear solutions related to this diagnostic method. I believe that the method should be completed by exemplifying the primers that can be used, but also the amplicons. One possible article that could help provide these details is: Nadăș, G.C., Kalmár, Z., Taulescu, M., ChirilÇŽ, F., Bouari, C.M., Răpuntean, S., Bolfă, P., & FiÈ›, N.I. (2014). Comparative identification of Candida species isolated from animals using phenotypic and PCR-RFLP methods. Bulletin of the Veterinary Institute in Pulawy, 58, 219 - 222.
Lines 234-244: Regarding antifungal therapy, I believe that some doses of antifungal substances that can be used in sloths should also be offered.
Line 242: Please add space before the new sentence
Lines 252-254: I think it would help to exemplify some classes of disinfectants
Line 258: Please add space before the sentence (Regular monitoring….)
Line 261: Please add space before the sentence (Combining these….)
Lines 316 – 319: Please use Italics for Xenarthra, Pilosa and Megalonychidae.
Line 317: Please add space after Choloepus didactylus [15]…
Lines 320 - 321: Please use Italics for Megalonychidae! Please delete “.” After “a).” and “b).”
Lines 332 – 358: I believe that the information provided in these paragraphs should be included in a subchapter. In its current form, it is presented together with general data on sloth species. For example 8.1. Species details, 8.2. Commonly reported diseases of sloths from captivity. So, 8.3 will be “Candidiasis in sloths”!
Line 452: Please add space after “.” And remove the space from “Trea tment”
Author Response
Dear Reviewer 1,
We are very grateful that you recognize the importance of this review. And we inform you that all changes, suggestions and requests for additional information were inserted in the text, and highlighted red. Thank you for all the suggestions, they certainly made this review better and more complete.
Sincerely,
Felipe Masiero Salvarani
Reviewer 2 Report
Comments and Suggestions for Authors
This is a very thorough review of Candida and how it relates to the two-toed sloth. There are only a few errors/ queries related to the general biology of Canidida
Section 2
lines 72,74 - ergosterol is a component of the plasma membrane of Candida, not the cell wall. Chitin, mannans, and glucans make up the cell wall of Candida.
line 73, 84 - it is considered that carbohydrates are the nutrient source for Candida and thus they actually produce CO2. Indeed high levels of CO2 have been found to inhibit growth. Perhaps this is different for Candida found in wild animals in environment - can the authors check this?
line 77 - some species of Candida are dimorphic, not all
4. line 176 - Pathogenesis is misspelled
line 176 - I don't think it is necessary to continually define Choleoepus - either species name or two-toed sloths is sufficient
6. line 255-264 - This section could be removed as quarantine is discussed in section 7 'prevention and control'
Author Response
Dear Reviewer 2,
We are very grateful that you recognize the importance of this review. And we inform you that all changes, suggestions and requests for additional information were inserted in the text, and highlighted red . Thank you for all the suggestions, they certainly made this review better and more complete.
Sincerely,
Felipe Masiero Salvarani